# Bandit Learning with Implicit Feedback

**Yi Qi[1], Qingyun Wu[2], Hongning Wang[2], Jie Tang[1], Maosong Sun[1]**
[1] State Key Lab of Intell. Tech. & Sys.,
Institution for Artificial Intelligence,
Dept. of Comp. Sci. & Tech., Tsinghua University, Beijing, China
[2] Department of Computer Science, University of Virginia
qi-y16@mails.tsinghua.edu.cn, {jietang, sms}@tsinghua.edu.cn
{qw2ky,hw5x}@virginia.edu

## Abstract

Implicit feedback, such as user clicks, although abundant in online information service systems, does not provide substantial evidence on users' evaluation of system's output. Without proper modeling, such incomplete supervision inevitably misleads model estimation, especially in a bandit learning setting where the feedback is acquired on the fly. In this work, we perform contextual bandit learning with implicit feedback by modeling the feedback as a composition of user result examination and relevance judgment. Since users' examination behavior is unobserved, we introduce latent variables to model it. We perform Thompson sampling on top of variational Bayesian inference for arm selection and model update. Our upper regret bound analysis of the proposed algorithm proves its feasibility of learning from implicit feedback in a bandit setting; and extensive empirical evaluations on click logs collected from a major MOOC platform further demonstrate its learning effectiveness in practice.

## 1   Introduction

Contextual bandit algorithms [4, 20, 19] provide modern information service systems an effective solution to adaptively find good mappings between available items and users. This family of algorithms sequentially select items to serve users using side information about user and item, while adapting their selection strategies based on the immediate user feedback to maximize users' long-term satisfaction. They have been popularly deployed in practical systems for content recommendation [20, 5, 26] and display advertising [6, 22].

However, the most dominant form of user feedback in such systems is implicit feedback, such as clicks, which is known to be biased and incomplete about users' evaluation of system's output [16, 11]. For example, a user skips a recommended item might not be because he/she does not like the item, but he/she just does not examine that display position, i.e., position bias [13]. Unfortunately, a common practice in contextual bandit applications simply treats no click as a form of negative feedback [20, 25, 6]. This introduces inconsistency to model update, since the skipped items might not be truly irrelevant, and it inevitably leads to suboptimal outputs of bandit algorithms over time.

In this work, we focus on learning contextual bandits with user click feedback, and model such implicit feedback as a composition of user result examination and relevance judgment. Examination hypothesis [8], which is a fundamental assumption in click modeling, postulates that a user clicks on a system's returned result *if and only if* that result has been examined by the user and it is relevant to the user's information need at the moment. Because a user's examination behavior is unobserved, we model it as a latent variable, and realize the examination hypothesis in a probabilistic model. We define the conditional probabilities of result examination and relevance judgment via logistic functions over the corresponding contextual features. To perform model update, we take a variational

Bayesian approach to develop a closed form approximation to the posterior distribution of model parameters on the fly. This approximation also paves the way for an efficient Thompson sampling strategy for arm selection in bandit learning. Our finite time analysis proves that, despite the increased complexity in parameter estimation introduced by the latent variables, our Thompson sampling policy based on the true posterior is guaranteed to achieve a sub-linear Bayesian regret with a high probability. We also demonstrate that the regret of Thompson sampling based on the approximated posterior is well-bounded. In addition, we prove that when one fails to model result examination in click feedback, a linearly increasing regret is possible, as the model cannot differentiate examination driven skips from relevance driven skips in the negative feedback.

We tested the algorithm in XuetangX[1], a major Massive Open Online Course (MOOC) platform in China, for personalized education. To personalize students' learning experience on this platform, we recommend quiz-like questions in a form of banners on top of the lecture videos when students are watching the videos. The algorithm needs to decide where in a video to display which question to a target student. If the student feels the displayed question is helpful for him/her to understand the lecture content, he/she could click on the banner to read the answer and more related online content about the question. Therefore, our goal is to maximize the click through rate (CTR) on the selected questions. There are several properties of this application that amplifies the bias and incompleteness of click feedback. First, based on the consideration of user experience, to minimize the risk of annoying any student, the displayed time of a banner is limited to a few seconds. Second, as this feature is newly introduced to the platform, many users might not realize that they can click on the question to read more related content about it. As a result, no click on a question does not necessarily indicate its irrelevance. We tested the algorithm in this application in a four-month period, where a total of 69 questions are manually compiled for the algorithm to select over 20 major videos with more than 100 thousands student video watching sessions. Based on the unbiased offline evaluation policy [21], our algorithm achieved a 8.9% CTR lift compared to standard contextual bandits [20, 9] which do not model users' examination behavior.

## 2 Related Works

As having been extensively studied in click modeling of user search results [7], various factors affect users' click decisions; and among them result examination plays a central role [13, 8]. Unfortunately, most applications of bandit algorithms simply treat user clicks as explicit feedback for model update [20, 25, 6, 26], where no click on a selected result is considered as negative feedback. This inevitably leads to inaccurate model update and sub-optimal arm selection. There is a line of related research that develops click model based bandit algorithms for learning to rank problems. For example, by assuming that skipped documents are less attractive than later clicked ones in a ranked list, Kveton et al. [17] developed a cascading bandit model to learn from both clicks and skips in search results. To enable learning from multiple clicks in the same result ranking list, they adopted the dependent click model [10] to infer user satisfaction after a sequence of clicks [14], and later further extended it to broader types of click models [27]. However, such algorithms aim at estimating the best ranking of results in a per-query basis, without specifying any specific ranking function. Hence, it is hard for them to generalize to unseen queries. This directly limits their application scenario in practice. The solution developed in Lagrée et al. [18] is the closest to ours, which exploits bias in reward distribution induced by different examination probabilities at different display positions. Yet they assumed the examination probability only depends on position, while we allow any reasonable feature to be a determinant. Besides, they postulated that the probability of examination at each position is either heuristically set or empirically estimated, and henceforth fixed; while we estimate it on the fly from the observations obtained by interacting with users.

Another line of related research is bandit learning with latent variables. Maillard and Mannor studied the problem of latent bandit [23], which assumes reward distributions are clustered and the clusters are determined by some latent variables. They only studied the problem in a context-free setting, and a very weak performance guarantee is provided when the reward distribution is unknown in those clusters. Kawale et al. developed a Thompson sampling scheme for online matrix-factorization [15]. Latent features are extracted via an online low-rank matrix completion based on samples selected from Thompson sampling on the fly. Due to the ad-hoc combination of factorization method and bandit method, little theoretical analysis was provided. Wang et al. studied the problem of latent

feature learning for contextual bandits [25]. They extended arms' context vectors with latent features under a linear reward structure, and applied the upper confidence bound principle over coordinate descent to iteratively estimate the hidden features and model parameters. The linear reward structure prohibits it from recognizing the nonlinear dependency between result examination and relevance judgment in click feedback. And their regret analysis depends heavily on the initialization of the algorithm, which could be hard to achieve in practice.

## 3 Problem Setup

We consider a contextual bandit problem with finite, but possibly large, number of arms. Denote the arm set as $\mathcal{A}$. At each trial $t = 1, ..., T$, the learner observes a subset of candidate arms $\mathcal{A}_t$ with $\mathcal{A}_t \subset \mathcal{A}$, where each arm $a$ is associated with a context vector $\mathbf{x}^a$ summarizing the side information about the arm. Once an arm $a_t \in \mathcal{A}_t$ is chosen according to some policy $\pi$, corresponding implicit binary feedback $C_{a_t}$, e.g., user click, will be given to the learner as the reward. The learner's goal is to adjust its arm selection strategy to maximize its cumulative reward over time. What makes this problem unique and challenging is that $C_{a_t}$ does not truly reflect users' evaluation of the selected arm $a_t$. Based on the examination hypothesis [13, 8], when $C_{a_t} = 1$, the chosen $a_t$ must be relevant to the user's information need at time $t$; but when $C_{a_t} = 0$, $a_t$ might be relevant but the user just does not examine it. Unfortunately, the result examination condition is unobserved to the learner.

We model a user's result examination via a binary latent variable $E_{a_t}$ and assume that the context vector $\mathbf{x}_t^a$ of arm $a$ can be decomposed into $(\mathbf{x}_{C,t}^a, \mathbf{x}_{E,t}^a)$, where the dimension of $\mathbf{x}_{C,t}^a$ and $\mathbf{x}_{E,t}^a$ are $d_C$ and $d_E$ respectively. Accordingly, users' result examination and relevance judgment decisions are assumed to be governed by a conjecture of $(\mathbf{x}_{C,t}^a, \mathbf{x}_{E,t}^a)$ and the corresponding bandit parameter $\boldsymbol{\theta}^* = (\boldsymbol{\theta}_C^*, \boldsymbol{\theta}_E^*)$. In the rest of this paper, when no ambiguity is introduced, we drop the index $a$ to simplify the notations. As a result, we make the following generative assumption about an observed click $C_t$ on arm $a_t$,

$$\mathbb{P}(C_t = 1 | E_t = 0, \mathbf{x}_{C,t}) = 0$$
$$\mathbb{P}(C_t = 1 | E_t = 1, \mathbf{x}_{C,t}) = \rho(\mathbf{x}_{C,t}^\mathsf{T} \boldsymbol{\theta}_C^*)$$
$$\mathbb{P}(E_t = 1 | \mathbf{x}_{E,t}) = \rho(\mathbf{x}_{E,t}^\mathsf{T} \boldsymbol{\theta}_E^*)$$

where $\rho(x) = \frac{1}{1+e^{-x}}$. Based on this assumption, we have $\mathbb{E}[C_t | \mathbf{x}_t] = \rho(\mathbf{x}_{C,t}^\mathsf{T} \boldsymbol{\theta}_C^*) \rho(\mathbf{x}_{E,t}^\mathsf{T} \boldsymbol{\theta}_E^*)$. As a result, the observed click feedback $C_t$ is a sample from this generative process. Define $f_{\boldsymbol{\theta}}(\mathbf{x}) := \mathbb{E}[C | \mathbf{x}, \boldsymbol{\theta}] = \rho(\mathbf{x}_C^\mathsf{T} \boldsymbol{\theta}_C) \rho(\mathbf{x}_E^\mathsf{T} \boldsymbol{\theta}_E)$. The accumulated regret of a policy $\pi$ up to time $T$ is formally defined as,

$$\text{Regret}(T, \pi, \boldsymbol{\theta}^*) = \sum_{t=1}^{T} \max_{a \in \mathcal{A}_t} f_{\boldsymbol{\theta}^*}(\mathbf{x}^a) - f_{\boldsymbol{\theta}^*}(\mathbf{x}^{a_t})$$

where $\mathbf{x}^{a_t} := (\mathbf{x}_C^{a_t}, \mathbf{x}_E^{a_t})$ is the context vector of the arm $a_t \in \mathcal{A}_t$ selected by the policy $\pi$ at time $t$ based on the history $\mathcal{H}_t := \{(\mathcal{A}_i, \mathbf{x}_i, C_i)\}_{i=1}^{t-1}$. The Bayesian regret is defined by $\mathbb{E}\big[\text{Regret}(T, \pi, \theta^*)\big]$, where the expectation is taken with respect to the prior distribution over $\boldsymbol{\theta}^*$, and it can be written as,

$$\text{BayesRegret}(T, \pi) = \sum_{t=1}^{T} \mathbb{E}\big[ \max_{a \in \mathcal{A}_t} f_{\boldsymbol{\theta}^*}(\mathbf{x}^a) - f_{\boldsymbol{\theta}^*}(\mathbf{x}^{a_t})\big]$$

In our online learning setting, the objective is to find the policy $\pi$ that minimizes the accumulated regret over T.

## 4 Algorithm

The learner needs to estimate the bandit parameters $\boldsymbol{\theta}_C^*$ and $\boldsymbol{\theta}_E^*$ based on its interactively obtained click feedback $\{\mathbf{x}_i, C_i\}_{i=1}^t$ over time. Ideally, this estimation can be obtained by maximizing the data likelihood with respect to the bandit model parameters. However, the inclusion of examination as a latent variable in our bandit learning setting poses serious challenges to both parameter estimation and arm selection. Neither conventional least square estimator nor maximum likelihood estimator can

be easily obtained, let alone computational efficiency, due to the non-convexity of the corresponding optimization problem. To make things even worse, the two popular bandit learning paradigms, upper confidence bound principle [1] and Thompson sampling [3], both demand an accurate estimation of bandit parameters and their uncertainty. In this section, we present an efficient new solution to tackle these two challenges, which makes use of variational Bayesian inference technique to learn parameters approximately on the fly, as well as to bridge parameter estimation and arm selection policy design.

## 4.1 Variational Bayesian for parameter estimation

To complete the generative process defined in Section 3, we further assume $\boldsymbol{\theta}_C$ and $\boldsymbol{\theta}_E$ follow Gaussian distribution $N(\hat{\boldsymbol{\theta}}_C, \boldsymbol{\Sigma}_C)$ and $N(\hat{\boldsymbol{\theta}}_E, \boldsymbol{\Sigma}_E)$ respectively. We are interested in developing a closed form approximation to their posteriors, when a newly obtained observation $(\mathbf{x}_C, \mathbf{x}_E, C)$ becomes available. By applying Bayes' rule in the log space, we have,

$$
\begin{aligned}
&\log \mathbb{P}(\boldsymbol{\theta}_C, \boldsymbol{\theta}_E | \mathbf{x}_C, \mathbf{x}_E, C) \\
&= \log \mathbb{P}(C | \boldsymbol{\theta}_C, \boldsymbol{\theta}_E, \mathbf{x}_C, \mathbf{x}_E) + \log \mathbb{P}(\boldsymbol{\theta}_C, \boldsymbol{\theta}_E) + \log \mathrm{const} \\
&= C \log \rho(\mathbf{x}_C^\mathsf{T} \boldsymbol{\theta}_C) \rho(\mathbf{x}_E^\mathsf{T} \boldsymbol{\theta}_E) + (1 - C) \log \left(1 - \rho(\mathbf{x}_C^\mathsf{T} \boldsymbol{\theta}_C) \rho(\mathbf{x}_E^\mathsf{T} \boldsymbol{\theta}_E)\right) \\
&\quad - \frac{1}{2}(\boldsymbol{\theta}_C - \hat{\boldsymbol{\theta}}_C)^\mathsf{T} \boldsymbol{\Sigma}_C^{-1} (\boldsymbol{\theta}_C - \hat{\boldsymbol{\theta}}_C) - \frac{1}{2}(\boldsymbol{\theta}_E - \hat{\boldsymbol{\theta}}_E)^\mathsf{T} \boldsymbol{\Sigma}_E^{-1} (\boldsymbol{\theta}_E - \hat{\boldsymbol{\theta}}_E) + \log \mathrm{const}
\end{aligned}
$$

The key idea is to develop a variational lower bound in the quadratic form of $\boldsymbol{\theta}_C$ and $\boldsymbol{\theta}_E$ for the log-likelihood function. Because of the convexity of $\log \rho(x) - \frac{x}{2}$ with respect to $x^2$ (See Appendix B.1) and the Jensen's inequality for $\log x$ (See Appendix B.2), a lower bound of the required form is achievable. When $C = 1$, by Eq (16) in Appendix B.3, we have,

$$
l_{C=1}(\mathbf{x}_C, \mathbf{x}_E, \boldsymbol{\theta}) := \log \left(\rho(\mathbf{x}_C^\mathsf{T} \boldsymbol{\theta}_C) \rho(\mathbf{x}_E^\mathsf{T} \boldsymbol{\theta}_E)\right) \geq g(\mathbf{x}_C^\mathsf{T} \boldsymbol{\theta}, \xi_C) + g(\mathbf{x}_E^\mathsf{T} \boldsymbol{\theta}, \xi_E) \tag{1}
$$

where $g(x, \xi) := \frac{x}{2} - \frac{\xi}{2} + \log \rho(\xi) - \lambda(\xi)(x^2 - \xi^2)$, $\lambda(\xi) = \frac{\tanh \frac{\xi}{2}}{4\xi}$, $x, \xi \in \mathcal{R}$. More specifically, $g(x, \xi)$ is a polynomial of degree 2 with respect to $x$. When $C = 0$, by Eq (17) in Appendix B.3, we have,

$$
\begin{aligned}
l_{C=0}(\mathbf{x}_C, \mathbf{x}_E, \boldsymbol{\theta}) &:= \log \left(1 - \rho(\mathbf{x}_C^\mathsf{T} \boldsymbol{\theta}_C) \rho(\mathbf{x}_E^\mathsf{T} \boldsymbol{\theta}_E)\right) \\
&\geq H(q) + q g(-\mathbf{x}_C^\mathsf{T} \boldsymbol{\theta}, \xi_C) + q g(\mathbf{x}_E^\mathsf{T} \boldsymbol{\theta}, \xi_{E,1}) + (1 - q) g(-\mathbf{x}_E^\mathsf{T} \boldsymbol{\theta}, \xi_{E,2})
\end{aligned} \tag{2}
$$

where $H(q) := -q \log q - (1 - q) \log(1 - q)$. Once the lower bound in the quadratic form is established, we can use a Gaussian distribution to approximate our target posterior, whose mean and covariance matrix are determined by the following equations,

$$
\boldsymbol{\Sigma}_{C,\mathrm{post}}^{-1} = \boldsymbol{\Sigma}_C^{-1} + 2q^{1-C} \lambda(\xi_C) \mathbf{x}_C \mathbf{x}_C^\mathsf{T} \tag{3}
$$

$$
\hat{\boldsymbol{\theta}}_{C,\mathrm{post}} = \boldsymbol{\Sigma}_{C,\mathrm{post}}(\boldsymbol{\Sigma}_C^{-1} \hat{\boldsymbol{\theta}}_C + \frac{1}{2}(-q)^{1-C} \mathbf{x}_C) \tag{4}
$$

$$
\boldsymbol{\Sigma}_{E,\mathrm{post}}^{-1} = \boldsymbol{\Sigma}_E^{-1} + 2\lambda(\xi_E) \mathbf{x}_E \mathbf{x}_E^\mathsf{T} \tag{5}
$$

$$
\hat{\boldsymbol{\theta}}_{E,\mathrm{post}} = \boldsymbol{\Sigma}_{E,\mathrm{post}}(\boldsymbol{\Sigma}_E^{-1} \hat{\boldsymbol{\theta}}_E + \frac{1}{2}(2q - 1)^{1-C} \mathbf{x}_E) \tag{6}
$$

where the subscript "post" denotes the parameters in the Gaussian distributions that approximate the desired posteriors. Consecutive observations can be incorporated into the approximated posteriors sequentially. There is one thing left to decide, i.e., the choice of variational parameters $(\xi_C, \xi_E, q)$. A typical criterion is to choose the values such that the likelihood on the observations is maximized. Similar to the choice made by [12], we choose the closed form update formulas of those variational parameters as,

$$
\xi_C = \sqrt{\mathbf{E}_{\boldsymbol{\theta}_C}[\mathbf{x}_C^\mathsf{T} \boldsymbol{\theta}_C]^2}
$$

$$
\xi_E = \sqrt{\mathbf{E}_{\boldsymbol{\theta}_E}[\mathbf{x}_E^\mathsf{T} \boldsymbol{\theta}_E]^2}
$$

$$
q = \frac{\exp \left(g(\mathbf{x}_C^\mathsf{T} \boldsymbol{\theta}_C, \xi_C) + g(\mathbf{x}_E^\mathsf{T} \boldsymbol{\theta}_E, \xi_E) - g(-\mathbf{x}_E^\mathsf{T} \boldsymbol{\theta}_E, \xi_E)\right)}{1 + \exp \left(g(\mathbf{x}_C^\mathsf{T} \boldsymbol{\theta}_C, \xi_C) + g(\mathbf{x}_E^\mathsf{T} \boldsymbol{\theta}_E, \xi_E) - g(-\mathbf{x}_E^\mathsf{T} \boldsymbol{\theta}_E, \xi_E)\right)}
$$

---

**Algorithm 1** Thompson sampling for E-C Bandit

---

1: Initiate $\boldsymbol{\Sigma}_C = \lambda I, \boldsymbol{\Sigma}_E = \lambda I, \hat{\boldsymbol{\theta}}_C = \boldsymbol{\theta}_{C,0}, \hat{\boldsymbol{\theta}}_E = \boldsymbol{\theta}_{E,0}$.
2: **for** $t = 0, 1, 2...$ **do**
3:     Observe the available arm set $\mathcal{A}_t \subset \mathcal{A}$ and its corresponding context set $\mathcal{X}_t := \{(\mathbf{x}_C^a, \mathbf{x}_E^a) : a \in \mathcal{A}_t\}$.
4:     Randomly sample $\tilde{\boldsymbol{\theta}}_C \sim N(\hat{\boldsymbol{\theta}}_C, \boldsymbol{\Sigma}_C), \tilde{\boldsymbol{\theta}}_E \sim N(\hat{\boldsymbol{\theta}}_E, \boldsymbol{\Sigma}_E)$.
5:     Select:
$$a_t = \arg\max_{a \in \mathcal{A}_t} \rho((\mathbf{x}_C^a)^\mathsf{T} \tilde{\boldsymbol{\theta}}_C) \rho((\mathbf{x}_E^a)^\mathsf{T} \tilde{\boldsymbol{\theta}}_E)$$
6:     Play the selected arm $a_t$ and Observe the reward $C_t$.
7:     Update $\boldsymbol{\Sigma}_C, \hat{\boldsymbol{\theta}}_C, \boldsymbol{\Sigma}_E, \hat{\boldsymbol{\theta}}_E$ according to Eq (3) to Eq (6) respectively.
8: **end for**

---

where all the expectations are taken under the approximated posteriors. Empirically, we found the iterative update of the approximated posterior and the variational parameters converge quite rapidly, such that it usually only needs a few rounds of iterations to get a satisfactory local maximum in our experiments.

## 4.2 Thompson sampling with approximated lower bound

Thompson sampling, also known as probability matching, is widely used in bandit learning to balance exploration and exploitation, and it shows great empirical performance [6]. Thompson sampling requires a distribution of the model parameters to sample from. In a standard Thompson sampling [3], one is required to sample from the true posterior of model parameters. But as logistic regression does not have a conjugate prior, the model defined in our problem does not have an exact posterior. We decide to sample from the approximated posterior as derived in Eq (3) to Eq (6). Later we will demonstrate this is a very tight posterior approximation. Once the sampling of $(\tilde{\boldsymbol{\theta}}_C, \tilde{\boldsymbol{\theta}}_E)$ is complete, we can select the corresponding arm $a_t \in \mathcal{A}_t$ which maximizes $\rho(\mathbf{x}_C^\mathsf{T} \tilde{\boldsymbol{\theta}}_C) \rho(\mathbf{x}_E^\mathsf{T} \tilde{\boldsymbol{\theta}}_E)$. We name the resulting bandit algorithm as examination-click bandit, or E-C Bandit in short, and summarize it in Algorithm 1.

## 5 Regret Analysis

Recall our object is to find the policy that minimizes the Beyesian regret,

$$\text{BayesRegret}(T, \pi) = \sum_{t=1}^{T} \mathbb{E}\Big[\max_{a \in \mathcal{A}_t} f_{\boldsymbol{\theta}^*}(\mathbf{x}^a) - f_{\boldsymbol{\theta}^*}(\mathbf{x}^{a_t})\Big]$$

where $f_{\boldsymbol{\theta}}(\mathbf{x}) := \mathbb{E}[C|\mathbf{x}, \boldsymbol{\theta}] = \rho(\mathbf{x}_C^\mathsf{T} \boldsymbol{\theta}_C) \rho(\mathbf{x}_E^\mathsf{T} \boldsymbol{\theta}_E)$. Our algorithm, which is based on a maximum likelihood estimator, is equivalent to an estimator that minimizes a log-loss with binary random variables. In this section, we will first bound the aggregate empirical discrepancy of the log-loss estimator used in our model in Proposition 1. This prepares for the upper bound of the generic Bayeisan regret under a log-loss estimator with Thompson sampling policy in Theorem 1. Based on this generic Bayesian regret bound, we study the upper bound of Bayesian regret for our proposed E-C Bandit. Due to space limit, we provide all the detailed proofs in the Appendix.

To further simplify our notations, we use $f$ for $f_{\boldsymbol{\theta}}$, which is the reward function based the estimated bandit parameter $\boldsymbol{\theta}$, and $f_k$ for $f_{\boldsymbol{\theta}}(\mathbf{x}^{a_k})$, i.e., the reward for arm $a_k$. We use $f^*$ for $f_{\boldsymbol{\theta}^*}$, which is the reward function based on the ground-truth bandit parameter, and correspondingly $f_k^*$ for $f_{\boldsymbol{\theta}^*}(\mathbf{x}^{a_k})$. We assume that $f^*$ lies in a known function space $\mathcal{F}$, where any $f \in \mathcal{F}$ is a function mapping from the arm set $\mathcal{A}$ to the range $(0, 1)$. Define the log-loss estimator by $\hat{f}_t^{\text{LOGLOSS}} \in \arg\min_{f \in \mathcal{F}} L_{2,t}(f)$ where $L_{2,t}(f)$ is the aggregate log-loss written as $\sum_{k=1}^{t-1} l_k(f)$ where $l_k(f) = -\big(C_k \log f_k + (1 - C_k) \log(1 - f_k)\big)$. We have the following proposition,

**Proposition 1.** Denote the aggregate empirical discrepancy $\sum_{k=1}^{t}(f_k - f_k^*)^2$ by $\|f - f^*\|_{E,t}^2$. For all $\delta > 0$ and $\alpha > 0$, if $\mathcal{F}_t = \left\{ f \in \mathcal{F} : \left\| f - \hat{f}_t^{\text{LOGLOSS}} \right\|_{E,t} \leq \sqrt{\beta_t^*(\mathcal{F}, \delta, \alpha)} \right\}$ for all $t \in N$,

$$\mathbb{P}\big(f^* \in \cap_{t=1}^{\infty} \mathcal{F}_t\big) > 1 - \delta, \tag{7}$$

where $\beta_t^*(\mathcal{F}, \delta, \alpha)$ is an appropriately constructed confidence parameter. In particular, it is defined as $\beta_t^*(\mathcal{F}, \delta, \alpha) := \frac{2}{\lambda_0} \log(N(\mathcal{F}, \alpha, \|\cdot\|_{\infty})/\delta) + 2\alpha\eta_t$, where $N(\mathcal{F}, \alpha, \|\cdot\|_{\infty})$ denotes the alpha-covering number of $\mathcal{F}$, $\lambda_0 = \frac{1}{(\frac{1}{m_f} + \frac{1}{1-M_f})^2}$ and $\eta_t = \big(4M_f + \frac{1}{\min\{m_f, 1-M_f\}}\big)t$, in which $m_f, M_f \in \mathcal{R}$ are upper and lower bounds of $f$ such at $0 < m_f \leq f \leq M_f < 1$ for any $f \in \mathcal{F}$.

**Remark 1.** The proof is provided in Appendix C. Here we discuss two important details related to our later proof about E-C Bandit's regret. First, the precise optimization of $\hat{f}_t^{\text{LOGLOSS}} \in \arg\min_{f \in \mathcal{F}} L_{2,t}(f)$ could be hard in some instances of $\mathcal{F}$. For example, when $\mathcal{F}$ is a set of non-convex functions. Nevertheless, we can always resort to approximation methods to solve the optimization problem as long as the approximation error can be bounded. Indeed, in our E-C Bandit, we resort to variational inference to estimate $\hat{f}_t^{\text{LOGLOSS}}$ on the fly and find it works quite well in practice. Second, when $f^* \notin \mathcal{F}$, this corresponds to the problem of model mis-specification. In this situation, the regret bound could be very poor, as the real regret could be linear with respect to time. To show this clearly in our case, in Appendix F we construct a situation in which the regret is inevitably linear if one fails to model the examination condition in click feedback and simply treats no click as negative feedback.

With Proposition 1, we have the following theorem which bounds the Bayesian regret of the Thompson Sampling strategy under a log-loss estimator.

**Theorem 1.** For all $T \in N$, $\alpha > 0$ and $\delta < \frac{1}{2T}$, if $\pi^{TS}$ denotes the policy derived from the log-loss estimator and a Thompson sampling strategy along the time steps, then

$$\text{BayesRegret}(T, \pi^{TS}) \leq 1 + \big( \dim_E^{\mathcal{A}}(\mathcal{F}, T^{-1}) + 1\big)C + 4\sqrt{\dim_E^{\mathcal{A}}(\mathcal{F}, T^{-1})\beta_T^*(\mathcal{F}, \alpha, \delta)T} \tag{8}$$

where $C = \sup_{f \in \mathcal{F}}\{\sup f\}$, $\dim_E^{\mathcal{A}}(\mathcal{F}, T^{-1})$ is the eluder dimension (see Definition 3 in Russo and Van Roy [24]) of $\mathcal{F}$ with respect to $\mathcal{A}$.

**Remark 2.** We can choose $C = 1$ in our click feedback case since $f \in (0, 1)$. $C$ is kept in the theorem to show the same form compared to the Proposition 8 in Russo and Van Roy [24]. In fact, the proof is almost the same once we have Proposition 1. Hence, we omit the proof in our paper.

Now we turn to provide an upper regret bound of our E-C Bandit, based on the above generic Bayeisan regret analysis under a log-loss estimator. We add the following two assumptions which are standard in the literature of contextual bandits.

**Assumption 1.** The optimal $\boldsymbol{\theta}^*$ lies in $\mathcal{B}_s := \{\boldsymbol{\theta} \in \mathcal{R}^d : \|\boldsymbol{\theta}\|_2 \leq s\}$, and $s$ is known as a prior.

**Assumption 2.** The norm of context vectors are bounded by $x$, i.e., $(\mathbf{x}_C, \mathbf{x}_E) \in \mathcal{B}_x$, where $\mathcal{B}_x := \{\mathbf{x} \in \mathcal{R}^d : \|\mathbf{x}\|_2 \leq x\}$ and $x$ is known as a prior.

Based on these two assumptions, it is straightforward to verify that $\rho(\mathbf{x}_C^\mathsf{T}\boldsymbol{\theta}_C), \rho(\mathbf{x}_E^\mathsf{T}\boldsymbol{\theta}_E)$ and $f_{\boldsymbol{\theta}}(\mathbf{x})$ are bounded. Let $M_\rho = \max_{\boldsymbol{\theta} \in \mathcal{B}_s, \mathbf{x} \in \mathcal{B}_x} \rho(\mathbf{x}^\mathsf{T}\boldsymbol{\theta})$ and $m_\rho = \min_{\boldsymbol{\theta} \in \mathcal{B}_s, \mathbf{x} \in \mathcal{B}_x} \rho(\mathbf{x}^\mathsf{T}\boldsymbol{\theta})$. Hence, $0 < m_\rho \leq M_\rho < 1$. Similarly, denote the maximum of $f_{\boldsymbol{\theta}}(\mathbf{x})$ by $M_f$ and the minimum by $m_f$, we have $0 < m_f \leq M_f < 1$. Once the arm set is restricted to a finite cardinality, we have $\dim_E^{\mathcal{A}}(\mathcal{F}, T^{-1}) \leq |\mathcal{A}|$ by Appendix C.1. in Russo and Van Roy [24]. Choosing the function class as that in our E-C Bandit, i.e., $\mathcal{F} = \{f : \mathcal{B}_x \to \mathcal{R}|f = \rho(\mathbf{x}_C^\mathsf{T}\boldsymbol{\theta}_C)\rho(\mathbf{x}_E^\mathsf{T}\boldsymbol{\theta}_E), \boldsymbol{\theta} \in \mathcal{B}_s\}$, by Lemma 8 (See Appendix 8 for its proof), we have $N(\mathcal{F}, \alpha, \|\cdot\|_{\infty}) = (\gamma/\alpha)^d$ where $\gamma = 2M_\rho k_\rho x$ ($k_\rho$ is the Lipschitz constant of $\rho$, see Lemma 4). Hence, choosing $\alpha = 1/t^2$ and $\delta = 1/t$ leads to

$$\beta_t^*(\mathcal{F}, 1/t, 1/t^2) = \frac{2}{\lambda_0}d\log(\gamma t^3) + \frac{1}{t}(4M_f + \frac{1}{m_f}). \tag{9}$$

Therefore, the upper bound of Bayesian regret of our proposed E-C Bandit takes the following form,

$$\text{BayesRegret}(T, \pi^{TS}) = O(|\mathcal{A}| + \sqrt{d|\mathcal{A}|T \log T}). \tag{10}$$

When $T \gg |\mathcal{A}|$ and $T \gg d$, which is a typical case in practice, it becomes $O(\sqrt{T \log T})$.

# 6 Experiment

We perform empirical evaluations in simulation and on the online student click logs collected from our MOOC platform to verify the effectiveness of our proposed algorithm. In particular, we compare with those that fail to model the examination condition and directly use click as feedback.

## 6.1 Algorithms for comparison

We list the models used for empirical comparisons below, and explain how we adjust them in our evaluations.

**Logistic Bandit.** This model has been extensively used for online advertisement CTR optimization. In [6, 21], the authors model user clicks by a regularized logistic regression model over observed context features and make decisions by applying Thompson sampling over the learnt model. In particular, no click is treated as negative feedback. Following their setting in [6], we used the Laplace approximation and Gaussian prior presented in to update the model parameters on the fly. We also want to highlight that despite mountains of works focusing on generalized linear bandits, most of them are not truly online algorithms, because the estimation of their parameters at each iteration has to involve all historical observations iteratively. This incurs a space complexity at least $O(T)$ and time complexity at least $O(T^2)$ (e.g., Filippi et al. [9] requires exact optimum of logistic regression on all historical observations at each round).

**hLinUCB.** This is an algorithm proposed by Wang et al. [25] for bandit learning with latent variables. It is related to our model in a sense that both models estimate hidden features. In particular, hLinUCB extends linear contextual bandit by inclusion of hidden features and operates under a UCB-like strategy. However, it still treats click as direct feedback, but aims at learning more expressive features to describe the observed clicks.

**E-C Bandit.** This is the algorithm we present in Algorithm 1. We should note that in the experiments on real-world data, the manual separation of examination feature $\mathbf{x}_E$ and click feature $\mathbf{x}_C$ in the context vector $\mathbf{x}$ offers a principled approach to incorporate one's domain knowledge about what affects user examination and what affects user result relevance. We explain in detail what features are chosen for which component in Appendix G. Thanks to the tight approximation achieved by Bayesian variational inference presented in Section 4, truly online model update is feasible in this algorithm. This provides both computational and storage efficiency.

## 6.2 Experiments on simulations

First we demonstrate the effectiveness of our algorithm by experiment with simulated data. The experiment is performed as follows. The context vector's dimension $d_C$ and $d_E$ are set to 5, and thus $d = d_C + d_E = 10$. We set $|\mathcal{A}| = 100$, each of which is associated with a unique context vector $(\mathbf{x}_C, \mathbf{x}_E)$. The ground-truth parameter $(\boldsymbol{\theta}_C^*, \boldsymbol{\theta}_E^*)$ and the specific value of $(\mathbf{x}_C, \mathbf{x}_E)$ are all randomly sampled from the unit ball $\mathcal{B} = \{\mathbf{x} \in \mathcal{R}^d : \|\mathbf{x}\|_2 \leq 1\}$. Since $(\boldsymbol{\theta}_C^*, \boldsymbol{\theta}_E^*)$ and $(\mathbf{x}_C, \mathbf{x}_E)$ are both sampled from $\mathcal{B}$, $m_f$ and $M_f$ can be obtained by taking the minimum and maximum of $\rho(\mathbf{x}_C^\mathsf{T} \boldsymbol{\theta}_C)\rho(\mathbf{x}_E^\mathsf{T} \boldsymbol{\theta}_E)$ on $\mathcal{B}$, respectively, i.e., $m_f = \frac{1}{(1+e)^2}$ and $M_f = \frac{1}{(1+e^{-1})^2}$.

At each time $t$, an arm set $\tilde{\mathcal{A}}_t$ is randomly sampled from $\mathcal{A}$ such that $|\tilde{\mathcal{A}}_t| = 10$, i.e., each time we offer 10 randomly selected arms for the algorithm to choose from. An algorithm selects an arm from $\mathcal{A}_t$ and observes the corresponding reward $C_t^{\mathrm{alg}}$ generated by the Bernoulli distribution $B(\rho(\mathbf{x}_{C,t}^\mathsf{T} \boldsymbol{\theta}_C^*)\rho(\mathbf{x}_{E,t}^\mathsf{T} \boldsymbol{\theta}_E^*))$. The regret of this algorithm at time $t$ is defined by its received click reward, i.e., $\mathrm{regret}(t) = C_{a_t^*} - C_{a_t}$, where $a_t^*$ is the optimum arm to be chosen based on the ground-truth bandit parameters $(\boldsymbol{\theta}_C^*, \boldsymbol{\theta}_E^*)$.

We repeat the experiment 100 times using the same simulation setting, each containing 10,000 iterations. The average cumulative regret over 100 runs and the corresponding standard deviation (plotted per thousand iterations) are illustrated in Figure 1. One can clearly notice that the Logistic Bandit suffers from a linear regret with respect to time $t$, as it mistakenly treats no click as negative feedback. Our E-C Bandit achieves a fast converging sub-linear regret. The result that hLinUCB performs the worst is expected, since it assumes a linear relation between click and context feature vectors. We further investigate how the aggregate empirical discrepancy between E-C bandit and

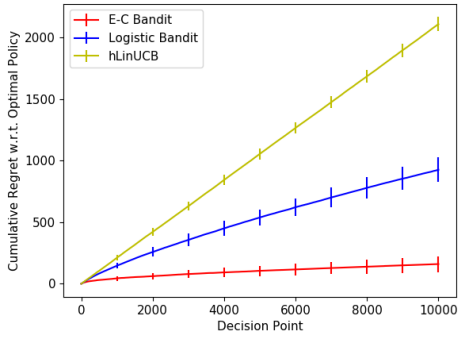

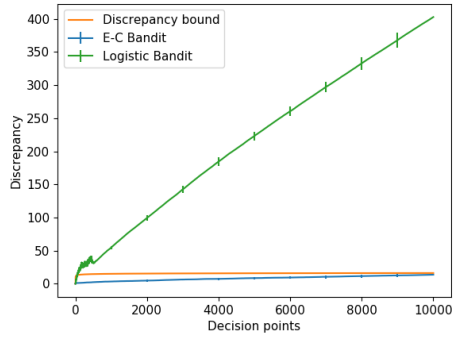

Figure 1: Comparison of cumulative regret over 100 runs of simulation.

Figure 2: Comparison of discrepancy bound provided by Proposition 1.

Logistic Bandit increases with respect to time. Figure 2 illustrates that the aggregate empirical discrepancy of E-C Bandit is well bounded by the upper bound provided by Proposition 1, while the Logistic Bandit's aggregate empirical discrepancy increases linearly. This directly explains their accumulative regret in this experiment comparison.

### 6.3 Experiments on MOOC video watching data

The MOOC data we used for evaluation is collected from a single course in a 4-month period. The course has 503 lecture videos in total. About 500 high-quality quiz-like questions have been manually crafted and each video is assigned with a subset of them based on human-judged relatedness. We selected 21 videos, whose accumulated watching time are ranked in the first 21 places, for this evaluation. Over the selected videos, on average a video is assigned with 5.5 questions, each of which is associated with 6 possible displaying positions within the video, leading to a total of 33 arms in average (as each question can be placed in all positions). The data set with our manually crafted features and our model implementation have been made publically available here: `https://github.com/qy7171/ec_bandit`.

We picked one video as an example to analyze students' click behavior. 9 arms are picked and projected by a random Gaussian matrix to a two-dimension plane in Figure 3. Thus, their relative distance are kept. The number in the parenthesis indicates the empirical CTR of the corresponding arm. It can be clearly seen that while arm $c$ and arm $f$ have the same empirical CTR, the arms between them, such as arm $a$ and $d$, have lower CTRs. Logistic Bandit is never able to capture this non-monotonicity relation, since its reward prediction increases monotonically with respect to a linear predictor. We construct a more general case in Appendix F to illustrate the scenario that failing to model examination condition would lead to a linear regret. Mapping the illustration back to the MOOC data set, arm $a$ and arm $f$ are two different questions displayed at the same position in the video, while arm $a$ and arm $c$ are the same question displayed at different positions. This phenomenon strongly suggests bias in users' implicit feedback, which again justifies our decomposition of examination and relevance in click feedback.

We followed [21] to develop our online data collection policy in our MOOC platform so as to prepare our offline evaluation data set. In particular, any related questions with respect to a video will have an equal probability to be selected and displayed at all positions in this video. We name this policy as $Similarity$. We create an instance of a bandit model for each video to learn its own optimal question placing policy. See Appendix G for detailed explanations of our examination feature choice. We also added a new baseline here, i.e., PBMUCB[18], which assumes a position-based examination probability in any ranking result. To adjust it to our setting, we assumed that the examination probability of any question chosen in a video is determined by and only by its position. Therefore, the key difference between our model and PBMUCB is that ours utilizes the available contextual information to estimate the examination probability, while PBMUCB is context-free. Yet, another important difference is that PBMUCB assumes the probability of examination at different position is known, and is estimated from offline data.

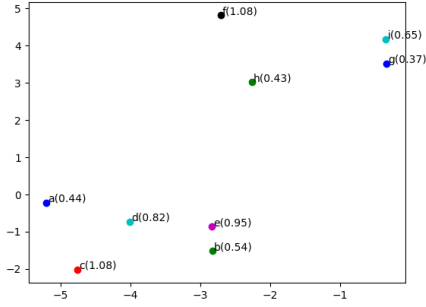

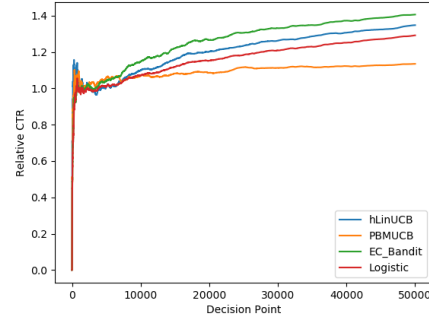

Figure 3: 9 arms' feature vectors projected onto a two-dimension plane, such that the relative distances between points are kept. The number in the parenthesis is the arm's empirical CTR.

Figure 4: Performance comparison on MOOC videos' data of different bandit algorithms.

Li et al. [21] proposed a method to calculate a near-unbiased estimate of CTR of any bandit algorithm based on the collected history data, so that offline evaluation and performance comparison are possible. We take their offline evaluation protocol here and report the estimated CTR in Figure 4, which is averaged over 100 runs. To avoid disclosing any proprietary information about the platform, all algorithms' CTRs are normalized by that from the $Similarity$ policy. As shown in the figure, independently estimating E-C Bandits across videos achieves an average $40.6\%$ increase in CTR over the $Similarity$ baseline. Meanwhile, E-C Bandit consistently outperforms the other three baseline bandits, i.e., hLinUCB, Logistic Bandit and PBMUCB. The improvement of our model compared to Logistic Bandit clearly suggests the necessity of modeling examination condition in user clicks for improving the online recommendation performance, and the improvement against PBMUCB provides strong evidence of the importance of modelling examination with available contextual information. In addition, the standard of error of E-C Bandit, hLinUCB, Logistic Bandit and PBMUCB's relative CTR performance among 100 trials are 0.032, 0.031, 0.030, 0.041, respectively. Therefore, the variance of our offline evaluation is small and the improvement from our solution to the baselines are statistically significant.

## 7 Conclusion

Motivated by the examination hypothesis in user click modeling, in this paper we developed E-C Bandit, which differentiates result examination and content relevance in user clicks and actively learns from such implicit feedback. We developed an efficient and effective learning algorithm based on variational inference and demonstrated its effectiveness on both simulated and real-world datasets. We proved that despite the complexity of underlying reward generation assumption and the resulting parameter estimation procedure, the proposed learning algorithm enjoys a sub-linear regret bound. Currently we only studied click feedback on single items; it is important for us to study it in a more general setting, e.g., a list of ranked items, where sequential result examination and relevance judgment introduce richer inter-dependency. In addition, our current regret analysis does not account for the additional discrepancy introduced by the variational inference. Abeille et al. [2] suggests that an exact posterior is not a necessary condition for a Thompson sampling policy to be optimal. It is important to study a tighter upper regret bound under our approximated posterior in general.

**Acknowledgements.** We thank the anonymous reviewers for their insightful comments. This paper is based upon work supported by a research fund from XuetangX.com and the National Science Foundation under grant IIS-1553568 and IIS-1618948.

## Footnotes

[1] http://www.xuetangx.com/

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
