[Supplementary Material]

## Appendix A  Preliminaries

In this section, we present some basic definitions and inequalities for later use.

First, let $p, q \in (0, 1)$, $H(p) := -p \log p - (1-p) \log(1-p)$ and $KL(p|q) := p \log \frac{p}{q} + (1-p) \log \frac{1-p}{1-q}$ denote the entropy and the KL-divergence function of Bernoulli variable respectively. Note that for the convenience of later derivations, we use natural logarithm here, which is slightly different from a standard definition.

The first inequality is Pinsker's inequality in Bernoulli case.

**Lemma 1** (Pinsker's inequality). $KL(p|q) - 2(p-q)^2 \geq 0$.

*Proof.* Define $c(q) := KL(p|q) - 2(p-q)^2$. We have $c'(q) = (p-q)(\frac{1}{q(1-q)} - 4)$. Since $q \in (0, 1)$, we have $q(1-q) \leq 1/4$ by the arithmetic average inequality. Thus, when $q < p, c'(q) <= 0$; when $q > p, c'(q) >= 0$. As a result, $c(q) \geq c(p) = 0$. $\square$

The next inequality provides a Lipschitz condition-like equality for $\log$ function defined on a bounded set.

**Lemma 2.** Given $a, b \in (0, 1)$, for any $x, y \in [a, b]$, $\frac{1}{b}|x - y| \leq |\log x - \log y| \leq \frac{1}{a}|x - y|$.

*Proof.* According to the mean value theorem, for any $x, y \in [a, b]$, there exists a $\eta \in [x, y]$ such that

$$\log x - \log y = \frac{1}{\eta}(x - y).$$

The inequality follows by taking the absolute value on two sides and the fact that $\eta$ is bounded in $[a, b]$. $\square$

The next is Hoeffding's lemma.

**Lemma 3** (Hoeffding's lemma). Let $X$ be any real-valued random variable such that $E(X) = 0$ and $a \leq X \leq b$ almost surely. Then, for all $\lambda \in \mathcal{R}$,

$$\mathbb{E}\big[e^{\lambda X}\big] \leq \lambda^2 (b-a)^2 / 8.$$

We omit its proof here, since its proof should be available in any standard textbook about the theory of probability. The last inequality deals with the Lipschitz condition of the logistic function.

**Lemma 4.** For the logistic function $\rho(x) = \frac{1}{1+e^{-x}}$, we have:

$$|\rho(x) - \rho(y)| < k_\rho |x - y|$$

where $k_\rho = 1/4$.

*Proof.* According to te mean value theorem, there exists a $\eta \in (x, y)$ such that

$$\rho(x) - \rho(y) = \rho(\eta)(1 - \rho(\eta))(x - y)$$

where we have used $\rho'(x) = \rho(x)(1 - \rho(x))$. Since $\rho(\eta) \in (0, 1)$, we have $\rho(\eta)(1 - \rho(\eta)) \leq 1/4$. Thus,

$$|\rho(x) - \rho(y)| = |\rho(\eta)(1 - \rho(\eta))| \times |(x - y)| \leq k_\rho |x - y|.$$

$\square$

## Appendix B  Variational lower bound for E-C Bandit

The variational lower bound in the quadratic form of our E-C bandit is constructed from two basic variational lower bounds. They are the lower bound of a logistic regression and the variational lower bound of a log-sum function. In this section, we first provide these two basic lower bounds in Section B.1 and Section B.2 respectively. We end this section by providing the desired lower bound for our E-C bandit in Section B.3.

## B.1 Variational lower bound for the log-logistic function

Jaakkola and Jordan [12] provide a variational lower bound for the log-logistic function, which takes the form of a 2-degree polynomial. This lower bound is based on the observation that $\log \rho(x) - \frac{x}{2}$ is convex with respect to $x^2$. We report their result here. Defining the related functions as follows:

$$\rho(x) := \frac{1}{1 + e^{-x}} (x \in \mathcal{R})$$

$$\lambda(\xi) := \frac{\tanh \frac{\xi}{2}}{4\xi}$$

$$g(x, \xi) := \frac{x}{2} - \frac{\xi}{2} + \log \rho(\xi) - \lambda(\xi)(x^2 - \xi^2)(x, \xi \in \mathcal{R})$$

We have the variational lower bound:

$$\log(\rho(x)) \geq g(x, \xi) \tag{11}$$

When $\xi^2 = x^2$, the lower bound is exact. Note that $1 - \rho(x) = \rho(-x)$. Thus:

$$\log(1 - \rho(x)) \geq g(-x, \xi) \tag{12}$$

The equation holds when $\xi^2 = x^2$.

## B.2 Variational lower bound for a log-sum function

We are interested in developing a bound for the following log-sum function, which is exactly the likelihood function of a single sample with reward $C = 0$ in E-C Bandit:

$$\log \left( (1 - \rho(\mathbf{x}_C^\mathsf{T} \boldsymbol{\theta}_C)) \rho(\mathbf{x}_E^\mathsf{T} \boldsymbol{\theta}_E) + 1 - \rho(\mathbf{x}_E^\mathsf{T} \boldsymbol{\theta}_E) \right). \tag{13}$$

Since $\log(x)$ is convex, we have the following inequality holds for any $q \in (0, 1)$:

$$\log \left( \left( (1 - \rho(\mathbf{x}_C^\mathsf{T} \boldsymbol{\theta}_C)) \rho(\mathbf{x}_E^\mathsf{T} \boldsymbol{\theta}_E) \right) + 1 - \rho(\mathbf{x}_E^\mathsf{T} \boldsymbol{\theta}_E) \right)$$
$$\geq H(q) + q \log \left( (1 - \rho(\mathbf{x}_C^\mathsf{T} \boldsymbol{\theta}_C)) \rho(\mathbf{x}_E^\mathsf{T} \boldsymbol{\theta}_E) \right) + (1 - q) \log \left( 1 - \rho(\mathbf{x}_E^\mathsf{T} \boldsymbol{\theta}_E) \right) \tag{14}$$

where $H(q) = -q \log q - (1 - q) \log(1 - q)$. It can be easily verified using Jensen's inequality for a convex function.

The equality holds whenever $q = \frac{\rho(\mathbf{x}_E^\mathsf{T} \boldsymbol{\theta}_E) - \rho(\mathbf{x}_E^\mathsf{T} \boldsymbol{\theta}_E) \rho(\mathbf{x}_C^\mathsf{T} \boldsymbol{\theta}_C)}{1 - \rho(\mathbf{x}_E^\mathsf{T} \boldsymbol{\theta}_E) \rho(\mathbf{x}_C^\mathsf{T} \boldsymbol{\theta}_C)}$, which always lies in the domain of $q$, i.e., $(0, 1)$, since $\rho(\mathbf{x}_C^\mathsf{T} \boldsymbol{\theta}_C) \in (0, 1)$ and $\rho(\mathbf{x}_E^\mathsf{T} \boldsymbol{\theta}_E) \in (0, 1)$.

## B.3 Variational lower bound for the log-likelihood of E-C Bandit

We provide the variational lower bound in a quadratic form for the log-likelihood of a single observation in E-C Bandit in this section. When $C = 1$, the log-likelihood of a single sample is:

$$l_{C=1}(\mathbf{x}_C, \mathbf{x}_E, \boldsymbol{\theta}_C, \boldsymbol{\theta}_E) = \log(\rho(\mathbf{x}_C^\mathsf{T} \boldsymbol{\theta}_C) \rho(\mathbf{x}_E^\mathsf{T} \boldsymbol{\theta}_E))$$

Since $\log(x)$ is additive, given Eq (11), we have:

$$l_{C=1}(\mathbf{x}_C, \mathbf{x}_E, \boldsymbol{\theta}_C, \boldsymbol{\theta}_E) = \log(\rho(\mathbf{x}_C^\mathsf{T} \boldsymbol{\theta}_C)) + \log(\rho(\mathbf{x}_E^\mathsf{T} \boldsymbol{\theta}_E)) \tag{15}$$
$$\geq g(\mathbf{x}_C^\mathsf{T} \boldsymbol{\theta}_C, \xi_C) + g(\mathbf{x}_E^\mathsf{T} \boldsymbol{\theta}_E, \xi_E)$$
$$=: \tilde{l}_{C=1}(\mathbf{x}_C, \mathbf{x}_E, \boldsymbol{\theta}_C, \boldsymbol{\theta}_E, \xi_C, \xi_E)$$

where $=:$ means "denoted by". When $C = 0$, the log-likelihood of a single sample is a log-sum function, which is the first lower-bounded by Eq (14). Then each log-logistic function is lower-bounded by Eq (11) and Eq (12). The derivation is as follows:

$$l_{C=0}(\mathbf{x}_C, \mathbf{x}_E, \boldsymbol{\theta}_C, \boldsymbol{\theta}_E) = \log(1 - \rho(\mathbf{x}_C^\mathsf{T} \boldsymbol{\theta}_C) \rho(\mathbf{x}_E^\mathsf{T} \boldsymbol{\theta}_E)) \tag{16}$$
$$= \log((1 - \rho(\mathbf{x}_C^\mathsf{T} \boldsymbol{\theta}_C)) \rho(\mathbf{x}_E^\mathsf{T} \boldsymbol{\theta}_E) + 1 - \rho(\mathbf{x}_E^\mathsf{T} \boldsymbol{\theta}_E))$$
$$\geq H(q) + q \log((1 - \rho(\mathbf{x}_C^\mathsf{T} \boldsymbol{\theta}_C)) \rho(\mathbf{x}_E^\mathsf{T} \boldsymbol{\theta}_E)) + (1 - q) \log(1 - \rho(\mathbf{x}_E^\mathsf{T} \boldsymbol{\theta}_E))$$
$$\geq H(q) + qg(-\mathbf{x}_C^\mathsf{T} \boldsymbol{\theta}_C, \xi_C) + qg(\mathbf{x}_E^\mathsf{T} \boldsymbol{\theta}_E, \xi_{E,1}) + (1 - q)g(-\mathbf{x}_E^\mathsf{T} \boldsymbol{\theta}_E, \xi_{E,2}) \tag{17}$$

where $\xi_C, \xi_{E,1}, \xi_{E,2}$ are the variational parameters of $\log(1 - \rho(\mathbf{x}_C^\mathsf{T}\boldsymbol{\theta}_C)), \log(\rho(\mathbf{x}_E^\mathsf{T}\boldsymbol{\theta}_E)), \log(1 - \rho(\mathbf{x}_E^\mathsf{T}\boldsymbol{\theta}_E))$ respectively. Note that the equality holds when $\xi_{E,1}^2 = (\mathbf{x}_E^\mathsf{T}\boldsymbol{\theta}_E)^2$ and $\xi_{E,2}^2 = (-\mathbf{x}_E^\mathsf{T}\boldsymbol{\theta}_E)^2$. We can always impose $\xi_{E,1} = \xi_{E,2}$ without harming the attainability of equality in the lower-bound. Thus, we can get a simplified lower-bound:

$$
\begin{aligned}
l_{C=0}(\mathbf{x}_C, \mathbf{x}_E, \boldsymbol{\theta}_C, \boldsymbol{\theta}_E) &\geq H(q) + qg(-\mathbf{x}_C^\mathsf{T}\boldsymbol{\theta}_C, \xi_C) + qg(\mathbf{x}_E^\mathsf{T}\boldsymbol{\theta}_E, \xi_E) + (1-q)g(-\mathbf{x}_E^\mathsf{T}\boldsymbol{\theta}_E, \xi_E) \\
&=: \tilde{l}_{C=0}(\mathbf{x}_C, \mathbf{x}_E, \boldsymbol{\theta}_C, \boldsymbol{\theta}_E, \xi_C, \xi_E, q)
\end{aligned}
$$

$$(18)$$

We can unify Eq (16) and Eq (18) as:

$$
\tilde{l}_C(\mathbf{x}_C, \mathbf{x}_E, \boldsymbol{\theta}_C, \boldsymbol{\theta}_E) := C\tilde{l}_{C=1}(\mathbf{x}_C, \mathbf{x}_E, \boldsymbol{\theta}_C, \boldsymbol{\theta}_E) + (1-C)\tilde{l}_{C=0}(\mathbf{x}_C, \mathbf{x}_E, \boldsymbol{\theta}_C, \boldsymbol{\theta}_E, \xi_C, \xi_E, q)
$$

which is a quadratic form of $(\boldsymbol{\theta}_C, \boldsymbol{\theta}_E)$ as desired, which can be easily verified by plugging in the definition of $g(x)$.

## Appendix C   Proof of confidence bound: Proposition 1

Proposition 1 is established by a union lower bound of aggregate empirical discrepancy of any function $f \in \mathcal{F}$ (in Lemma 5) with respect to the optimal $f^*$ and a discretization error bound based on an $\alpha$-covering of the function space $\mathcal{F}$ (in Lemma 6). In this section, we first present the proofs of these two lemmas, and end this section by a proof of Proposition 1 based on the two lemmas.

### C.1   Bound of log-loss estimator by aggregate empirical discrepancy

We lower bound the log-loss of any function $f \in \mathcal{F}$ in terms of the empirical log-loss of the true function $f^*$ and the aggregate empirical discrepancy $\|f - f^*\|_{E,t}^2 := \sum_{i=1}^t (f_k - f_k^*)^2$. The result is stated as the following lemma.

**Lemma 5.** For all $\delta > 0$ and $f \in \mathcal{F}$, with probability at least $1 - \delta$,

$$
\mathbb{P}\left(L_{2,t+1}(f) \geq L_{2,t+1}(f^*) + \frac{1}{2}\|f - f^*\|_{E,t}^2 - \frac{1}{\lambda_0}\log\frac{1}{\delta}\right) \geq 1 - \delta \tag{19}
$$

holds simultaneously for all natural number $t \in N$, where $\lambda_0 := \frac{3}{(\frac{1}{m_f} + \frac{1}{1-M_f})^2}$.

We first provide some definitions and properties that are helpful here. Let $\mathcal{H}_t^x = \mathcal{H}_t \cup \{\mathcal{A}_t, \mathbf{x}_t\}$ be a new filtration. First, $C_k \in \{0,1\}$ is a binary variable whose expectation conditioned on $\mathcal{H}_k^x$ is $f_k^*$. Hence, the random variable $\zeta_k := C_k - f_k^*$ is sub-Gaussian according to Lemma 7 such that:

$$
\mathbb{E}\left[\zeta_k | \mathcal{H}_k^x\right] = 0
$$

$$
\mathbb{E}\left[e^{\lambda\zeta_k} | \mathcal{H}_k^x\right] \leq \frac{\lambda^2}{2}
$$

The log-loss over a single sample in E-C Bandit is defined as $l_k(f) = -\left(C_k \log f_k + (1-C_k)\log(1-f_k)\right)$. The KL-divergence of two binary variables whose expectation are $p$ and $q$ respectively is defined by $KL(p|q) := p\log\frac{p}{q} + (1-p)\log\frac{1-p}{1-q}$, which satisfies Pinsker's Inequality (Lemma 1):

$$
KL(p|q) \geq 2(p-q)^2
$$

*Proof.* We first add the following definitions to simplify the notations:

$$
\begin{aligned}
h_k &:= \log f_k - \log f_k^* + \log(1 - f_k^*) - \log(1 - f_k) \\
Z_k &:= l_k(f^*) - l_k(f) \\
\Phi_k(\lambda) &:= \log\mathbb{E}\left[e^{\lambda(Z_t - E(Z_t))} | \mathcal{H}_k^x\right].
\end{aligned}
$$

By definition, $\sum_{i=1}^t Z_k = L_{2,t+1}(f^*) - L_{2,t+1}(f)$. Plugging in the equalities $C_k = \zeta_k + f_k^*$ and $l_k(f) = -\left(C_k \log f_k + (1-C_k)\log(1-f_k)\right)$, we can get that:

$$
Z_k = h_k\zeta_k - KL(f_k^*|f_k). \tag{20}
$$

Therefore, $\mathbb{E}\big[Z_k|\mathcal{H}_k^x\big] = -KL(f_k^*|f_k)$ and $\Phi_k(\lambda) = \log \mathbb{E}\big[e^{\lambda h_k \zeta_k}|\mathcal{H}_k^x\big]$. Hence, by Martingale Exponential Inequality (See Appendix B.1. in Russo and Van Roy [24]), for all $x \geq 0, \lambda \geq 0$,

$$\mathbb{P}\left(\lambda \sum_{i=1}^{t} Z_k \leq x - \lambda \sum_{i=1}^{t} KL(f_k^*|f_k) + \sum_{i=1}^{t} \log \mathbb{E}\big[e^{\lambda h_k \zeta_k}|\mathcal{H}_k^x\big]\right) \geq 1 - e^{-x}. \quad (21)$$

Since $KL(p|q) \geq 2(p-q)^2$ according to Lemma 1, and $\mathbb{E}\big[e^{\lambda \zeta_k}|\mathcal{H}_k^x\big] \leq \frac{\lambda^2}{2}$ according to Lemma 7, we have,

$$\mathbb{P}\left(\lambda \sum_{i=1}^{t} Z_k \leq x - 2\lambda \sum_{i=1}^{t} (f_k - f_k^*)^2 + \sum_{i=1}^{t} \lambda^2 h_k^2/2\right) \geq 1 - e^{-x}. \quad (22)$$

Again, according to Lemma 2, $|h_k| \leq |\log f_k - \log f_k^*| + |\log(1 - f_k^*) - \log(1 - f_k)| \leq (\frac{1}{m_f} + \frac{1}{1-M_f})|f_k - f_k^*|$, we have the following,

$$\mathbb{P}\left(\lambda \sum_{i=1}^{t} Z_k \leq x - 2\lambda \sum_{i=1}^{t} (f_k - f_k^*)^2 + \sum_{i=1}^{t} \lambda^2 (\frac{1}{m_f} + \frac{1}{1-M_f})^2 (f_k - f_k^*)^2/2\right) \geq 1 - e^{-x}. \quad (23)$$

Plugging in the definition of aggregate empirical discrepancy and $Z_t$, and rearranging the terms we get,

$$\mathbb{P}\left(L_{2,t+1}(f^*) - L_{2,t+1}(f) \leq \frac{x}{\lambda} + ((\frac{1}{m_f} + \frac{1}{1-M_f})^2 \frac{\lambda}{2} - 2)\|f - f^*\|_{E,t}^2\right) \geq 1 - e^{-x}. \quad (24)$$

By choosing $\lambda = \lambda_0 := \frac{3}{(\frac{1}{m_f} + \frac{1}{1-M_f})^2}$ and $x := \log \frac{1}{\delta}$, we have,

$$\mathbb{P}\left(L_{2,t+1}(f) \geq L_{2,t+1}(f^*) + \frac{1}{2}\|f - f^*\|_{E,t}^2 - \frac{1}{\lambda_0} \log \frac{1}{\delta}\right) \geq 1 - \delta. \quad (25)$$

Note that $\lambda_0 > 0$ thus satisfies the assumption of Martingale Exponential Inequality's assumption. Also note that the deduction above makes no assumption about the value of $t \in N$, thus the lemma is proved. $\qquad\square$

## C.2 Discretization error

**Lemma 6.** If $f^\alpha$ satisfies $\|f - f^\alpha\|_\infty \leq \alpha$, and $m_f > 0, M_f < 1$, we have

$$\left|\frac{1}{2}\|f^\alpha - f^*\|_{E,t}^2 - \frac{1}{2}\|f - f^*\|_{E,t}^2 + L_{2,t}(f) - L_{2,t}(f^\alpha)\right| \leq \alpha \eta_t \quad (26)$$

where $\eta_t := (4M_f + \frac{1}{\min\{m_f, 1-M_f\}})t$.

*Proof.* It is sufficient to consider a single sample's discretization error and then sum over all samples over time $t$. Thus we omit the subscript $t$ in the following derivation,

$$\begin{aligned}
\left|(f^\alpha - f^*)^2 - (f - f^*)^2\right| &= |(f^\alpha - f)(f^\alpha + f) + 2f^*(f - f^\alpha)| \quad (27) \\
&\leq |(f^\alpha - f)(f^\alpha + f)| + |2f^*(f - f^\alpha)| \\
&\leq 2M_f\alpha + 2M_f\alpha \\
&= 4M_f\alpha
\end{aligned}$$

and

$$\begin{aligned}
|l(f) - l(f^\alpha)| &= |C(\log f^\alpha - \log f) + (1 - C)(\log(1 - f^\alpha) - \log(1 - f))| \quad (28) \\
&\leq C|(\log f^\alpha - \log f)| + |(1 - C)(\log(1 - f^\alpha) - \log(1 - f))| \\
&\leq \frac{C}{m_f}|f^\alpha - f| + \frac{1 - C}{1 - M_f}|f^\alpha - f| \\
&\leq \frac{1}{\min\{m_f, 1 - M_f\}}\alpha
\end{aligned}$$

Summing over $t$ and using triangular inequality, we conclude the proof. $\qquad\square$

## C.3 Proof of Proposition 1

*Proof.* Let $\mathcal{F}^\alpha$ be an $\alpha$-covering of $\mathcal{F}$ with respect to the sup norm such that for any $f \in \mathcal{F}$, there exists an $f^\alpha \in \mathcal{F}^\alpha$ satisfies $\|f - f^\alpha\|_\infty \leq \alpha$. By a union bound of Lemma 5, with probability at least $1 - \delta$,

$$L_{2,t}(f^\alpha) - L_{2,t}(f^*) \geq \frac{1}{2}\|f^\alpha - f^*\|_{E,t}^2 - \frac{1}{\lambda_0}\log(|\mathcal{F}|^\alpha/\delta) \ \forall t \in N, f^\alpha \in \mathcal{F}^\alpha.$$

Thus, with probability at least $1 - \delta$ for all $t \in N$ and $f \in \mathcal{F}$,

$$L_{2,t}(f) - L_{2,t}(f^*) \geq \frac{1}{2}\|f - f^*\|_{E,t}^2 - \frac{1}{\lambda_0}\log(|\mathcal{F}^\alpha|/\delta) \tag{29}$$

$$+ \underbrace{\min_{f^\alpha \in \mathcal{F}^\alpha}\left\{\frac{1}{2}\|f^\alpha - f^*\|_{E,t}^2 - \frac{1}{2}\|f - f^*\|_{E,t}^2 + L_{2,t}(f) - L_{2,t}(f^\alpha)\right\}}_{\text{Discretization error}}$$

Since $\hat{f}_t^{\text{LOGLOSS}} \in \arg\min_{f \in \mathcal{F}} L_{2,t}(f)$ and $f^* \in \mathcal{F}$, we have

$$L_{2,t}(\hat{f}_t^{\text{LOGLOSS}}) - L_{2,t}(f^*) \leq 0. \tag{30}$$

Using the two-side bound of the dicretization error term established in Lemma 6, we find with probability at least $1 - \delta$,

$$\frac{1}{2}\left\|\hat{f}_t^{\text{LOGLOSS}} - f^*\right\|_{E,t}^2 \leq \frac{1}{\lambda_0}\log(|\mathcal{F}|^\alpha/\delta) + \alpha\eta_t \tag{31}$$

Taking the infimum over the size of $\alpha$-covers implies,

$$\left\|f^* - \hat{f}_t^{\text{LOGLOSS}}\right\|_{E,t} \leq \sqrt{\beta_t^*(\mathcal{F}, \delta, \alpha)}$$

where $\beta_t^*(\mathcal{F}, \delta, \alpha) := \frac{2}{\lambda_0}\log(N(\mathcal{F}, \alpha, \|\cdot\|_\infty)/\delta) + 2\alpha\eta_t$. $\qquad\square$

# Appendix D  The sub-Gaussian property of $C_t - f_t^*$

In this section, we prove $C_t - f_t^*$ is a sub-Gaussian random variable.

**Lemma 7.** Let $\mathcal{H}_t^x = \mathcal{H}_t \cup \{\mathcal{A}_t, \mathbf{x}_t\}$ be a new filtration constructed on top of $\mathcal{H}_t$, where $\mathcal{H}_t := \{(\mathcal{A}_i, \mathbf{x}_i, C_i)\}_{i=1}^{t-1}$ and $C_t$ is a binary random variable such that $\mathbb{E}[C_t|\mathbf{x}_t] = f_t^*$. The random variable $\zeta_t := C_t - f_t^*$ is $\sigma$-sub-Gaussian conditioned on $\mathcal{H}_t^x$. Therefore, we have

$$\mathbb{E}[\zeta_t|\mathcal{H}_t^x] = 0$$

and there exists a $\sigma > 0$ such that

$$\mathbb{E}[e^{\lambda\zeta_t}|\mathcal{H}_t^x] \leq e^{\frac{\lambda^2\sigma^2}{2}}$$

for any $\lambda \in \mathcal{R}$.

*Proof.* Since $\mathbb{E}[C_t|\mathbf{x}_t] = f_t^*$ and $f_t^* \in \mathcal{H}_t^x$, we have $\mathbb{E}[\zeta_t|\mathcal{H}_t^x] = \mathbb{E}[C_t - f_t^*|\mathcal{H}_t^x] = 0$. Since $\zeta_t$ is bounded in $(-1, 1)$, by Hoeffding's lemma,

$$\mathbb{E}[e^{\lambda\zeta_t}|\mathcal{H}_t^x] \leq e^{\frac{\lambda^2}{2}}.$$

Thus conditioned on $\mathcal{H}_t^x$, $\zeta_t$ is $\sigma$-sub-Gaussian, where one can choose $\sigma = 1$. $\qquad\square$

# Appendix E  Alpha-covering number of function space

We provide an $\alpha$-covering number for the function space used in E-C Bandit.

**Lemma 8.** Let $\boldsymbol{\theta} := [\boldsymbol{\theta}_C^\mathsf{T}, \boldsymbol{\theta}_E^\mathsf{T}]^\mathsf{T} \in \mathcal{B}_s$, $\mathbf{x} := [\mathbf{x}_C^\mathsf{T}, \mathbf{x}_E^\mathsf{T}]^\mathsf{T} \in \mathcal{B}_x$, where $\mathcal{B}_s := \{\boldsymbol{\theta} \in \mathcal{R}^d : \|\boldsymbol{\theta}\|_2 \leq s\}$ and $\mathcal{B}_x := \{\mathbf{x} \in \mathcal{R}^d : \|\mathbf{x}\|_2 \leq x\}$. Let $\mathcal{F} = \{f : \mathcal{B}_x \to \mathcal{R} | f = \rho(\mathbf{x}_C^\mathsf{T}\boldsymbol{\theta}_C)\rho(\mathbf{x}_E^\mathsf{T}\boldsymbol{\theta}_E), [\boldsymbol{\theta}_C^\mathsf{T}, \boldsymbol{\theta}_E^\mathsf{T}]^\mathsf{T} \in \mathcal{B}_s\}$. Let $\gamma = 2M_\rho k_\rho x$. We have that

$$N(\mathcal{F}, \alpha, \|\cdot\|_\infty) = (\gamma/\alpha)^d$$

holds for any $\alpha > 0$.

*Proof.* For any $\boldsymbol{\theta}_1, \boldsymbol{\theta}_2 \in \mathcal{B}_s$, we have

$$\|f(\boldsymbol{\theta}_1) - f(\boldsymbol{\theta}_2)\|$$
$$= \left\|\rho(\mathbf{x}_C^\mathsf{T}\boldsymbol{\theta}_{C,1})\rho(\mathbf{x}_E^\mathsf{T}\boldsymbol{\theta}_{E,1}) - \rho(\mathbf{x}_C^\mathsf{T}\boldsymbol{\theta}_{C,1})\rho(\mathbf{x}_E^\mathsf{T}\boldsymbol{\theta}_{E,2}) + \rho(\mathbf{x}_C^\mathsf{T}\boldsymbol{\theta}_{C,1})\rho(\mathbf{x}_E^\mathsf{T}\boldsymbol{\theta}_{E,2}) - \rho(\mathbf{x}_C^\mathsf{T}\boldsymbol{\theta}_{C,2})\rho(\mathbf{x}_E^\mathsf{T}\boldsymbol{\theta}_{E,2})\right\|$$
$$\leq \left\|\rho(\mathbf{x}_C^\mathsf{T}\boldsymbol{\theta}_{C,1})\rho(\mathbf{x}_E^\mathsf{T}\boldsymbol{\theta}_{E,1}) - \rho(\mathbf{x}_C^\mathsf{T}\boldsymbol{\theta}_{C,1})\rho(\mathbf{x}_E^\mathsf{T}\boldsymbol{\theta}_{E,2})\right\| + \left\|\rho(\mathbf{x}_C^\mathsf{T}\boldsymbol{\theta}_{C,1})\rho(\mathbf{x}_E^\mathsf{T}\boldsymbol{\theta}_{E,2}) - \rho(\mathbf{x}_C^\mathsf{T}\boldsymbol{\theta}_{C,2})\rho(\mathbf{x}_E^\mathsf{T}\boldsymbol{\theta}_{E,2})\right\|$$
$$\leq M_\rho(\left\|\rho(\mathbf{x}_E^\mathsf{T}\boldsymbol{\theta}_{E,1}) - \rho(\mathbf{x}_E^\mathsf{T}\boldsymbol{\theta}_{E,2})\right\| + \left\|\rho(\mathbf{x}_C^\mathsf{T}\boldsymbol{\theta}_{C,1}) - \rho(\mathbf{x}_C^\mathsf{T}\boldsymbol{\theta}_{C,2})\right\|)$$
$$\leq M_\rho k_\rho\left\|\mathbf{x}_E^\mathsf{T}(\boldsymbol{\theta}_{E,1} - \boldsymbol{\theta}_{E,2})\right\| + \left\|\mathbf{x}_C^\mathsf{T}(\boldsymbol{\theta}_{C,1} - \boldsymbol{\theta}_{C,2})\right\|$$
$$\leq 2M_\rho k_\rho x\|\boldsymbol{\theta}_1 - \boldsymbol{\theta}_2\|$$
$$= \gamma\|\boldsymbol{\theta}_1 - \boldsymbol{\theta}_2\|$$

where all norms are infinity norm. Thus an $\alpha$-covering of $\mathcal{F}$ can be achieved by a $(\alpha/\gamma)$-covering of $\mathcal{B}_s$. Evenly divide $\mathcal{B}_s$ in each dimension, we prove that $N(\mathcal{F}, \alpha, \|\cdot\|_\infty) = (\gamma/\alpha)^d$, where we omit the ceiling function for simplicity. $\square$

## Appendix F   Linear regret of a mis-specified model

Figure 5: A contour plot of $f(x,y) = \rho(-x)\rho(-2y)$. Four arms are labeled by $A, B, C, D$ such that the four points forms a convex quadrilateral and both pairs of opposite vertexes have the same function value (reward). One can always find such four points due to the non-convexity of $f$.

In this section, we construct an example when a Logistic Bandit suffers from a linear regret under the examination's hypothesis. It is sufficient to consider a simple case with $d_C = d_E = 1$ and $|\mathcal{A}| = 4$. Figure 5 plots a simple but representative case when $\boldsymbol{\theta} = [1,2]^T$. One can think of the $x$-axis as the context feature for result examination and $y$-axis as the context feature for relevance judgment. Denote the reward of arm $a$ by $R_a$, and the corresponding coordinate by $(x_a, y_a)$, where $a \in \{A, B, C, D\}$. Then the figure indicates that

$$R_A = R_C > R_B = R_D. \tag{32}$$

Note that for a Logistic Bandit, which does not consider the decomposition of implicit feedback thus combines $x$ and $y$ in a linear predictor, will not be able to learn the correct order between those arms, i.e., Eq (32). This can be easily verified through the following necessary condition. In fact, assume there is $[a, b]^T$ such that

$$
\begin{aligned}
ax_A + by_A &= ax_C + by_C \\
ax_B + by_B &= ax_D + by_D
\end{aligned}
\tag{33}
$$

which is a necessary condition for a Logistic Bandit to learn Eq (32) exactly. As long as $x_B < x_A < x_C < x_D$ and $y_A < y_B < y_D < y_C$ (the case we plot in the figure), the determinant of coefficient matrix is less than 0 so that the only solution is $a = b = 0$. In this case, Logistic Bandit fails to differentiate $A, C$ from $B, D$, and would suffer from a regret of $R_A - R_B$ with probability $\frac{1}{2}$ when ties break arbitrarily. Without loss of generality, we assume $a \neq 0$. If $a > 0, b \geq 0$, then $ax_D + by_D > ax_A + by_A$, thus a regret of $R_A - R_D$ is inevitable when the availble arm set is $\{A, D\}$, since logistic function is monotonically increasing with respect to its linear predictor. So is the case when $a < 0, b \leq 0$ with the inequality $ax_B + by_B > ax_C + by_C$. The cases when $a > 0, b < 0$ with the inequality $ax_D + by_D > ax_C + by_D$, and when $a < 0, b > 0$ with the inequality $ax_B + by_B > ax_A + by_A$ follow the same argument. As a result, we construct a situation with an adversary environment when a linear regret is inevitable for a Logistic Bandit, as it cannot correctly model the implicit feedback.

## Appendix G    Features used for modeling of click and examination

We list the features we used for the modeling result examination and click in E-C Bandit in Table 1. Table 2 gives detailed description of these features. The main criteria of separating features between examination and relevance is whether this feature is question-related or not.

Table 1: Feature usage in E-C Bandit

| Feature | Used for examination? | Used for click(relevance)? |
|---|---|---|
| Hour in the Day | $\sqrt{}$ | $\times$ |
| Device | $\sqrt{}$ | $\times$ |
| Position | $\sqrt{}$ | $\times$ |
| Content Related | $\times$ | $\sqrt{}$ |

Table 2: Feature description

| Features | Description |
|---|---|
| Hour in the Day | In different hour, students are assumed to have different examination pattern. For example, when late at night, examination is less likely compared to daytime. |
| Device | A student may watch a video using different devices, such as a PC, iPad or smart cellphones. Different device should have different probability of examination. |
| Position | The question may be displayed at the beginning of the video, or somewhere middle of the video. Different CTRs for the same question at different positions are observed in real data, which we assume is caused by examination. |
| Content Related | A question's semantic meaning and its similarity with the subtitle of a video. Distributed embedding or tf-idf score may be used to gain a numeric representation of these features. |