[Reviews · NeurIPS 2018]

Reviewer 1



The paper considers an important and interesting problem of learning when "no-click" in user response does not necessarily mean a negative feedback, as the user may not have paid attention to the recommendation. The authors propose a latent variable model to capture this. They use Thompson Sampling (for logistic regression) with variational inference method for parameter estimation in order to handle the latent variables and obtain approximate posterior. Bayesian regret bounds are provided. I think the paper is conceptually very interesting and attacks an important problem, with a general technique that may have implications beyond the current setting. The regret bounds are not clean, but that may be expected in such a setting. The paper provides a good start for work in this important direction. Post rebuttal: I did not have any pressing concerns for the paper. I read the authors' response to the questions posed by other reviewers. I still like the paper as much as I liked it before and my score is unchanged.

Reviewer 2



The paper introduce ECBandit, a new algorithm for the bandit problem where the feedback is biased by the display bias. The method is based on variational bayesian inference and relies on some latent variables that account for the 'attention' given by the user (feedback provider) to the action played by the bandit. Experiments on synthetic and proprietary datasets are provided. - At high level, it would be useful to clarify why a variant of the position-based models often employed in ranking algorithms cannot be adapted to this case. - While the problem is of practical interest, the main experiment reported in the paper does not seem to be reproducible. Regarding the same experiment, the number of baselines tested is extremely small and it is really hard to understand. - While the algorithm samples several parameters (step4), the experiments do not do have plot with error bars

Reviewer 3



Summary: This work considers learning user preferences using a bandit model. The reward is not only based on the judgement of the user, but also whether the user examined the arm. That is feedback = examination * judgement In particular, if a user does not examine an arm, lack of feedback does not necessarily indicate that the user does not "like" the arm. This work uses a latent model for the (unobserved) examination of arms, and posits that the probability of positive feedback (binary) can be expressed as a product of the probability of examination (logistic) and positive feedback (logistic). The work proposes a VI approach to estimating the parameters, and then use a Thompson Sampling approach from the approximate posterior as policy. This allows them to use machinery from Russo and Van Roy to obtain regret bounds. Furthermore, simulations on synthetic and real data from a MOOC platform are provided. Specific comments: - The specific form of the model (product of 2 logistic regressions) is not really motivated in the problem formulation section. Why is this the relevant model as compared to other models? How can the reader tell whether this model is also relevant to their problem? - The objective is not given before the algorithm section. It would make sense to me to state the objective in the problem formulation section. - Will code / data be made available for others to replicate results? - Line 169: I assume you mean argmin instead of argmax. - The regret bounds from Russo and Van Roy require sampling from the true posterior, but this paper considers an approximation. Thus, the regret bound does not hold for the proposed algorithm. Authors could be a bit more clear about this. - On synthetic data, the EC algorithm beats benchmarks by a wide margin, but on the MOOC data this difference is much smaller. This suggests that there might important aspects that are not modeled by the current setup. Of course one cannot expect results on synthetic data to carry over completely, but maybe the authors can comment on this difference. - This is a more high level comment: the paper considers a model where the platform decides what action to take / show, and the user gives feedback. However, there are many scenarios where the platform provides a set of actions, of which the user takes one (e.g. a recommendation system). In the MOOC setting considered in the paper, the model in the paper is definitely the correct one, but it is not clear to me whether this generalizes to many other scenarios as well, or that in most cases the user selects actions rather than the platform. In the paper this distinction between the two viewpoints is somewhat blurred. It would be useful to hear the authors thoughts and for the work to be more precise on which settings are and are not relevant to the model presented in the paper. - Results in figure 4 seem very noisy. What are the SEs on the performances and could variance be reduced by running more replications? Overall: The work tackles a realistic problem, is well written and easy to follow. Theoretical results rely on standard techniques developed earlier. THe work is missing some motivation for the specific form of the model. The simulations are not wholy convincing, since the lift for the MOOC data is much smaller than that in the synthetic data. On the other hand, using real data is a big plus and proves that the approach can be used in practice. All in all, I'm on the fence for this one and look forward to the authors response to the issues raised -- Update after author response: I agree with other reviewers that it is interesting model and nice application of variational inference as well. I am a bit disappointed in the author's responses considering my comment re motivation of the specific model: I understand that the model is the product of two logistic functions, but would love to see more motivation on why this model over other alternatives, and/or how this model can be validated. Overall I remain of the opinion that it is a nice paper that covers an end-to-end use case with theory and practice. On the other hand, results are not groundbreaking so my score of a marginal accept stands.

Reviewer 4



Summary: This work study a generalized linear bandit model that decouples the estimation of the CTR into a contextual and an examination part. The authors derives a Thompson Sampling based on a variational lower bound of the posterior. TL;DR: Good paper but related work to be polished General comment: I think this work presents a very relevant model for click-through rate estimation. The algorithm builds smartly on TS and they are able to provide an analysis for it as well as a solid experimental section. Overall, it is a good paper but I am concerned by the related work section. Little work has been done to link this work with existing models. In particular, the present click model is rather close to the Position-Based Model (PBM) than to the Cascading Model studied by Kveton et al. [16]. The PBM has been widely studied and would deserve some comments. Please, add such references in a future version of the paper. This is a late review so I won't ask questions that the authors cannot answer. My only concern regarding the model is that in this setting, the features have to be somewhat arbitrarily split by the practitioner according to wether they encode rather context or examination. This has possibly an important impact on the final result. I wonder a bit how this is done in the MOOC experiment, it is not clear from the text.